# Low-Voltage, High-Frequency Synchronous Motor for Aerospace Applications

**Daniel Matt** [1,*]**, Lorenzo Piscini** [2]**, Nadhem Boubaker** [3]**, Anthony Gimeno** [4]**, Philippe Enrici** [1] 
**and Mourad Aitakkache** [1]

1 Institut d'Electronique et des Systèmes, University of Montpellier, 34095 Montpellier, France
2 Groupe Valeo, 38070 Saint-Quentin-Fallavier, France
3 Safran Electrical & Power, Pitstone LU7 9GT, UK
4 Technocentre Renault, 78280 Guyancourt, France
* Correspondence: daniel.matt@umontpellier.fr

**Abstract:** This article details the design of a permanent magnet synchronous electric motor prototype dedicated to the direct drive of the propeller for VTOL (Vertical Take-Off and Landing) and CTOL (Conventional Take-off and Landing) aircrafts. Our main aim is to maximise the power-to-weight ratio whilst not compromising the efficiency and the reliability. The originality of the research is based on the implementation of an armature winding using solid copper bars; we show that it is possible to use such an approach in an electric machine operating at very high frequency (1800 Hz) through a precise study on the shape of the bars to counter the additional losses. A prototype has been successfully manufactured; manufacturing details and some of the experimental test results are presented here.

**Keywords:** high power density; high frequency; electric aircraft; VTOL; low voltage electric motor; synchronous machine; permanent magnet; solid bar winding; power splitting

## 1. Introduction

The electric vehicle is now on the rise, which presents an ongoing technical and societal revolution. The pace of change is accelerating in a way that many western countries have challenged the reluctance of the automotive industries and strongly committed to weaning themselves away from fossil fuel engines (ICE engines) by pledging for zero-emission electric cars. The same transition is looming large in the horizon of the aerospace industry, where the future of flying seems to be irrevocably hybrid electric and full-electric.

Large all-electric commercial aircraft might not become viable in the near future due to, amongst other factors, the limited specific energy of battery technologies. However, small-scale full and hybrid electric aircrafts are under development and tens of projects are ongoing around the world, either based on new developed platforms or retrofits of existing ones. These small e-aircrafts can be classified into two mains categories: Conventional Take-off and Landing (eCTOLs), dedicated to different purposes, such as training, recreational flying clubs, etc., and Vertical Take-off and Landing (eVTOLs), which can be simply described as large drones for urban air mobility (UAM). Some examples of these light aircrafts from different ongoing projects are given in Figure 1.

The mass of the electric motor used for the propulsion of those aircrafts is a fundamental factor; it is not negligible compared to that of the structure or the source of energy (battery, etc.). Different solutions are therefore sought to maximise the power-to-weight ratio without comprising the efficiency and the reliability. There are however a number of technical barriers which limit this specific power; in a traditional design based on copper, iron alloy, and magnets, it is admitted that this limit is around 10 kW/kg for the considered rotational speeds.

We present in this article some original electromagnetic concepts for permanent magnet electric motors based on solid bar winding offering an attractive performance in terms of power-to-weight ratio. Furthermore, these concepts are using high pole count rotor in order to drastically reduce the size of the magnetic circuits (i.e., stator back iron and rotor yoke). The increase of the power density automatically leads to a higher loss density, and therefore particular attention shall be paid to the cooling method of the motor.

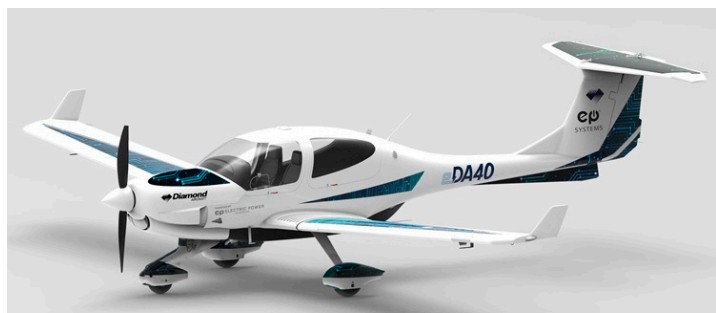

(**a**) eDA40 Diamond aircraft—All-Electric single engine trainer

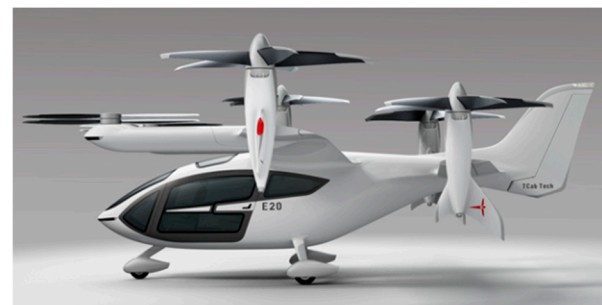

(**b**) E20 eVTOL UAM aircraft

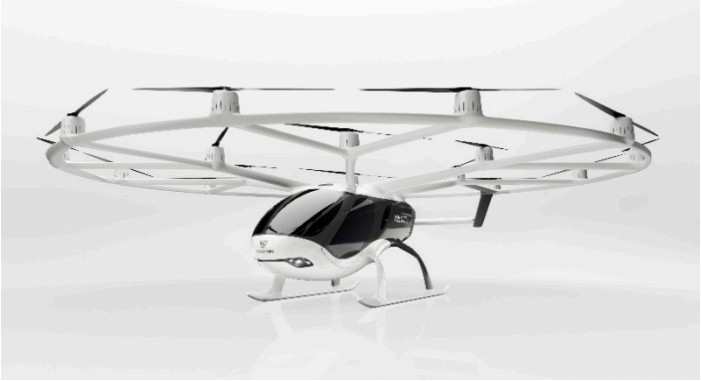

(**c**) Volocopter eVTOL UAM aircraft

**Figure 1.** Different concepts and architectures of full-electric aeroplanes.

The problems to be solved in this approach have been known for a long time, in particular the phenomenon of increased losses at high frequency in the conductors due to the current pushback into the slot. The literature, even old, is abundant on this subject [1–10]. The presented work is completely original, as it shows how to optimally use the concept of massive bars in order to considerably increase the copper filling rate of the slots while controlling the losses. This article is the synthesis of a PhD thesis [1], which completes similar works [2,3] by going further in the implementation of high frequencies and by proposing more successful constructive solutions.

This research work has been conducted in partnership with Safran Group, who provided the technical specifications requirement presented in the Table 1.

**Table 1.** Technical specifications requirement of the electric motor.

| Parameter | Value |
| --- | --- |
| Maximum continuous power @ 2500 RPM | 42 kW |
| Nominal output torque | 160 Nm |
| Transient power @ 2800 RPM (for max duration of 30 s) | 60 kW |
| VDC voltage supply (battery) | 300 VDC |
| Maximum overall outer diameter including housing fins | 290 mm |
| Maximum rotor inertia | 0.03 kg m$^2$ |
| Maximum length of the wound stator core pack | 75 mm |
| External air flow for cooling | 200 L/s |
| Minimum efficiency | 92% |

## 2. Description of the Developed Concept

### 2.1. Solid Bar Winding

The most original aspect of the motor concept presented here lies in the new topology of the stator winding. Conventionally, the winding of an electric motor is made from round copper wires, as illustrated in Figure 2b. However, when the VDC supply voltage is low, the number of parallel branches in the winding increases and/or the number of turns per coil decreases. In this case, it would be more judicious to rather opt for a solid copper conductor, commonly called a "hairpin bar", as illustrated in Figure 3.

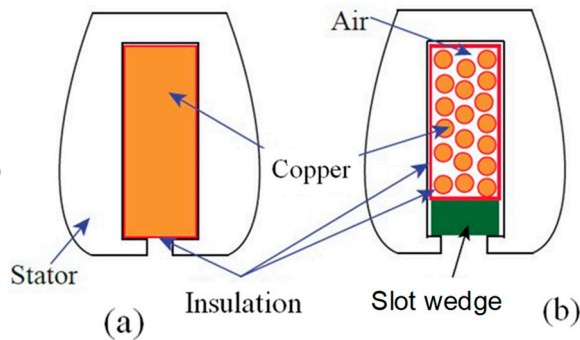

**Figure 2.** (**a**) Solid bar winding vs. (**b**) round wire winding.

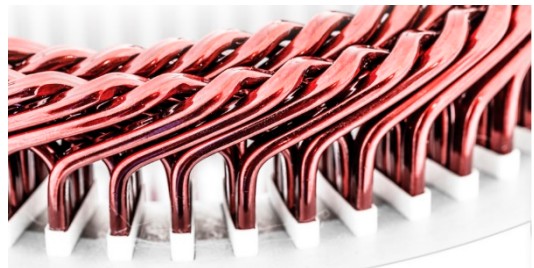

**Figure 3.** Hairpin winding (Courtesy of Special Machine Tool Company).

When appropriate conditions of the supply voltage are met, it would be even more beneficial, as it will be demonstrated here, to replace the round wire or hairpin bar by a single solid conductor in the slot, as shown in Figure 2a.

In summary, the main advantages of using one solid bar per slot are:

- High copper fill factor of 80%, instead of 30 to 40% maximum with a conventional winding.
- The iron-copper thermal resistance is reduced.
- The slot's opening width can be very narrow (approximately 0.5 mm) resulting in the increase of the flux density in the air gap (more torque output), decrease of the cogging torque/torque ripple, and reduction of AC copper loss due to the rotating magnets.
- The copper overhangs are very compact and well-controlled; this aspect will be further developed in this article.
- The winding manufacturing process is simplified and can be easily automated.
- The machine is more robust and reliable by tremendously reducing the likelihood of the occurrence of short-circuits between phases.

However, the use of solid bar conductors in the winding causes some constraints that need to be addressed, namely:

- The solution does not work for a low number of slots unless the motor is operating at a very low voltage.
- The solution does not benefit from the already well-established manufacturing processes and requires new tools and processes.

- The main drawback is the excessive loss due to the nature of the solid conductor that is more prone to the AC copper loss. The latter can be fully controlled if the phenomena causing the losses are well-understood. These phenomena are well-studied in the bibliographic references [1–10], where the main principles will be recalled here without getting into details.

In order to quantify the loss increase, the $K_{AC}$ coefficient is introduced, which is the ratio of the total AC copper loss, $P_{AC}$, to the DC copper loss (ohmic loss), $P_{DC}$, in the winding at given current:

$$K_{AC} = P_{AC}/P_{DC} \qquad (1)$$

The phenomenon causing excess copper loss in a solid conductor is known as the field effect or inductance effect. Unlike the conventional skin effect, it only takes place in the copper volume surrounded by a magnetic material (e.g., stator). Indeed, the additional copper loss is due to the transverse flux (slot leakage flux) produced by the main current, which closes through the slot and creates induced currents in the solid bar (eddy-currents) which will lead to an uneven current density distribution, being much higher in the lowermost part (near the slot opening) than in the uppermost part of the solid bar. This phenomenon is depicted in Figure 4.

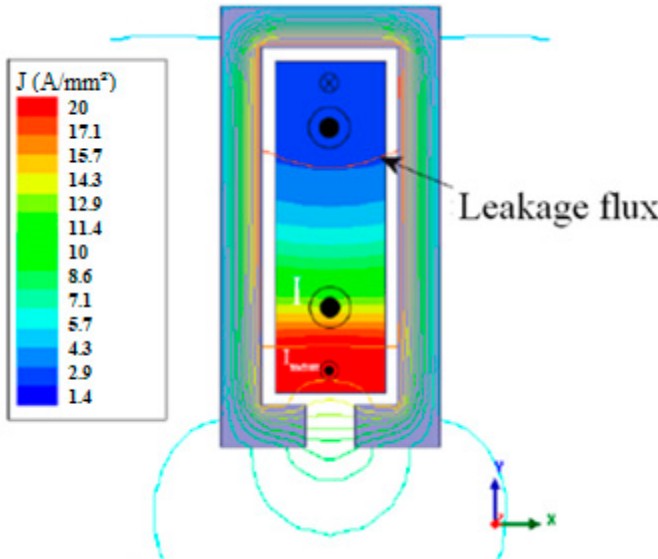

**Figure 4.** Illustration of the uneven distribution of the current density inside the conductor due to the slot transverse flux [2].

The coefficient $K_{AC}$ related the field effect could be precisely calculated using the following analytical relationship [2,3,7], which is based on the slot dimensions presented in Figure 5:

$$K_{AC} = \frac{h_{bar}}{\delta}\sqrt{\frac{t_{bar}}{t_{enc}}} \qquad (2)$$

In the above relationship, $\delta$ represents the skin depth in the copper, and the relationship is only valid when $h_{bar} > \delta$; otherwise, $K_{AC}$ is almost equal to 1.

The relationship (2) can be considered in order to optimally size the solid conductor. At an operating frequency of around 1800 Hz, which is our target here, the skin depth at 100 °C copper temperature is approximately equal to 1.8 mm. Thus, by selecting a bar height, $h_{bar}$, of 2 mm, the achieved $K_{AC}$ coefficient is still very close to 1, and the motor can operate at high frequency without causing excessive copper loss.

The bar width, $t_{bar}$, is then adjusted in order to get the right cross section of the copper bar needed to carry the high current. This leads to a topology having large and short slots.

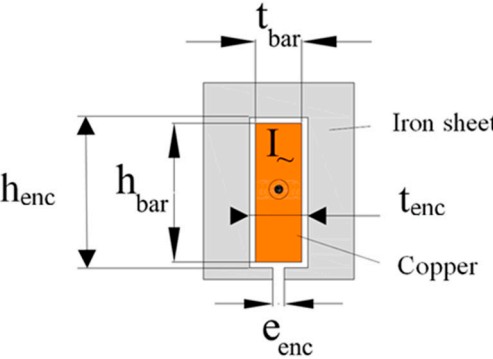

**Figure 5.** Main dimensions of the slot and the copper conductor.

Another similar approach based on the same analytical modelisation consists of considering the optimum height of the bar, which is called the critic height, $h_{critic}$. At given current, frequency and bar width, the losses tend to stagnate (or increase very slightly) when the bar height exceeds the $h_{critic}$, and this despite the increase of the conductor cross-section. The critic height can be calculated using the following equation [8]:

$$h_{critic} = \alpha \cdot \delta \cdot \sqrt{\frac{t_{enc}}{t_{bar}}} \qquad (3)$$

where $\alpha$ is a constant coefficient equal to 1.32 for the case of one bar per slot.

The above analytical relationships (2) and (3) are interesting because they help to easily grasp the essence of the complex phenomena at the origin of the excess copper loss; their accuracy has been corroborated by finite elements analysis considering the concrete case of the electric motor described in this article [1–3].

For motors operating at constant speed, the bar can be easily sized and optimised for the frequency corresponding to the constant rotational speed. However, for example, in the case of an electric vehicle where the motor has to operate at a more dynamic duty cycle, simultaneously requiring high torque/low speed, high torque/high speed (transient acceleration phase), and low torque/high speed, the sizing of the bar can be more delicate, where the cross section shall be chosen as a trade-off between the high frequency at high speed (AC loss) and the high current at high torque regime (DC loss).

For the motor described in this article, the nominal speed is almost constant and equal to 2500 rpm, as quoted in Table 1.

### 2.2. General Architecture of the Motor and Winding Selection

According to the technical specifications requirement, the motor shall be totally enclosed and cooled via the external fins of the housing by the forced airflow from the propeller of the aircraft. Accordingly, it has been decided that the optimum topology for the application would be an inner runner radial flux PM motor, where the majority of the losses are concentrated in the stator core pack and in the copper armature winding, therefore close to the cooling source. The thermal analysis is presented below.

In order to maximise the power-to-weight ratio, the fundamental frequency of the motor has been pushed as high as possible. The frequency was not set as an optimisation criterion but rather as an arbitrary target to be achieved while considering the manufacturability of the stator laminations [2]. The frequency target was chosen as 1800 Hz, which results in a number of poles of approximately 88 at 2500 rpm rotational speed. This choice necessarily dictates a fractional number of slots per pole and per phase (q < 1) so that the slot width, $t_{enc}$, can remain compatible with the solid bar winding solution, which requires a certain degree of freedom for the slot width (rather than the height) in order to limit the AC loss in the copper.

For the sake of limiting the overlapping between the phases in the end-winding, and hence, simplifying the manufacturing process of the winding with solid conductors, we opted for a winding layout where the conductors of each phase are grouped into an independent sector of adjacent slots. It is a concentrated winding around the tooth where the number of slots is very close to the number of poles. Many configurations are possible based on this winding layout (i.e., $N_{slot} = 2p \pm 1$ or $2p \pm 2$); an illustrative example of 24 slots/22 poles is given in Figure 6.

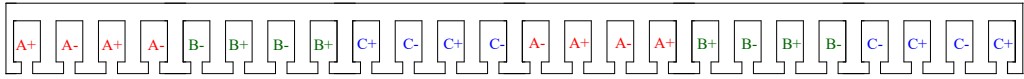

**Figure 6.** Concentrated winding, 22 poles/24 slots, where the phases are grouped into independent sectors.

For our prototype, we rather opted for a non-conventional winding [1,2,11], as illustrated in Figure 7, where the layout is still very similar to the conventional one described above. However, the main difference lies in introducing intermediate teeth to obtain the 120° electric shift between the phases while achieving a perfect alignment between the stator slots and the rotor poles, therefore maximising the flux linkage. In other words, contrary to the conventional layout, the stator slot pitch becomes equal to the rotor pole pitch, where all the elementary back EMFs within each group (phase) are in phase. Accordingly, this new winding offers a unit winding factor (kw = 1), which is impossible to achieve with the conventional layout (e.g., winding factor of the conventional case presented in Figure 6 is 0.958).

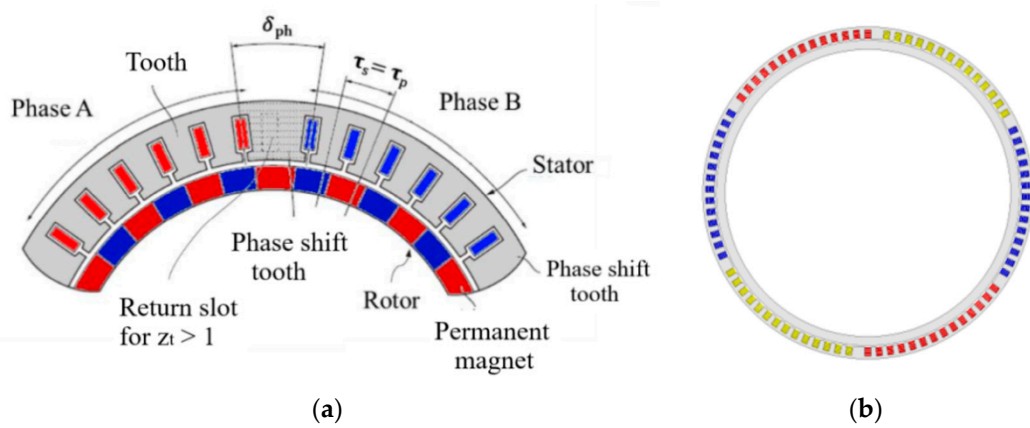

(**a**)　　　　　　　　　　　　　　　　　　　　　　　　　(**b**)

**Figure 7.** Non-conventional concentrated winding. (**a**) Detailed view. (**b**) Overview of the complete stator.

As illustrated in Figure 7b, the winding is symmetrical, where each phase is distributed over two diametrically opposed sectors, thereby reducing the bearings load and the amount of vibration and noise. Furthermore, when needed, this winding topology lends itself well to be configured into two mechanically and magnetically well decoupled stars, hence providing two separate power channels that can share the total load (reduces the inverter ratings) and act independently of one another, which is useful for achieving full redundancy and fault-tolerance capabilities, where one channel is able to continue operating at full power if the other power channel fails. The latter aspects are paramount in the context of VTOL or CTOL applications.

In the end, the electrical phase shift between two adjacent groups (phases), $\delta_{ph}$ (cf. Figure 7), was taken to be equal to 300° mainly in order to simplify the manufacturing of the double layer winding (two conductors per slot), as we will see in Section 3. Following the rules governing the design of this non-conventional winding and considering the targeted frequency of 1800 Hz, the number of poles (2p) and the number of slots shall be equal to 88 and 84 respectively, thereby each group (phase) encompasses 14 slots.

### 3. Design/Sizing of the Prototype and Material Selection

*3.1. Electromagnetic Design*

The VDC supply voltage and the output power are defined by the specifications in Table 1. Assuming double star winding (described earlier) and following the simple design logic described in [1], the phase current shall be equal to:

$$I_{ph} = 200 \ A_{rms} \tag{4}$$

The phase current is not an optimisation parameter, but rather indirectly imposed by the supply voltage available at the motor terminals; thus, the current shall be adjusted in order to achieve the required torque.

The airgap surface, $S_a$, is equal to the product of the circumference of the stator bore, $2 \cdot \pi \cdot R_s$, and the stator active length, $L_s$. The data from Table 1 enable the calculation $S_a$ assuming $L_s \approx 50mm$ and $R_s \approx 100$; the later are rough estimations before any optimisation:

$$S_a = 2 \cdot \pi \cdot R_s \cdot L_s = 0.03 \ m^2 \tag{5}$$

Thus, the shear stress in the airgap, $P_a$, calculated from the torque, $T_n$, is equal to:

$$P_a = T_n/(R_s \cdot S_a) \approx 5000 \ daN/m^2 \tag{6}$$

The average shear stress produced by the interaction between the stator currents, $I_{ph}$, and the amplitude of airgap flux density from the magnets, $B_m$, can be expressed as follows:

$$P_a = n_c \cdot \sqrt{2} \cdot I_{ph} \cdot n_s \cdot B_m/(4 \cdot \pi \cdot R_s) \tag{7}$$

where $n_c$ is the number of conductors per slot and $n_s$ is the number of slots. With one conductor per slot, 84 slots, and considering $B_m = 1.2$ T, the obtained shear stress is only 2400 daN/m$^2$, which is almost 50% of the required value (Equation (6)) in order to achieve 160 Nm output torque (requirement Table 1). Given the fact the envelope of the motor (length) is imposed, the only degree of freedom is to double the number of conductors per slot using double layer winding (i.e., $n_c = 2$); Figure 8 shows the complete winding.

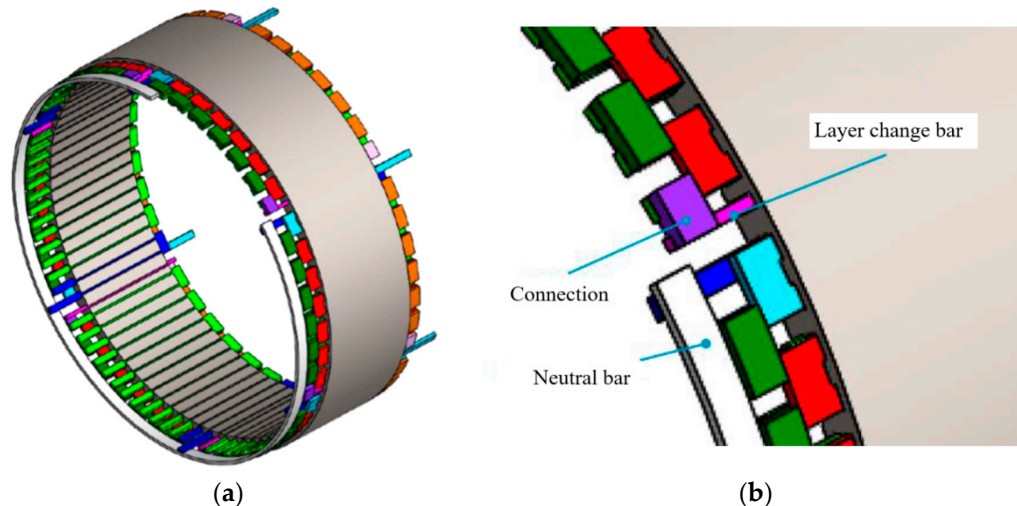

(**a**) (**b**)

**Figure 8.** Double layer non-conventional winding based on solid bar conductor: (**a**) overview, (**b**) detailed view.

The brazing connections between the bars at the end-winding is performed in two stages in the axial direction of the motor, which results in longer winding overhands. The return conductor for the second layer has been located in the intermediate big tooth

between two adjacent groups of the winding; therefore, the intermediate tooth becomes active and participates to the creating of the torque.

The optimum height (i.e., critic height) of the bar can always be calculated using the Equation (3). Doubling the number of conductors (two bars) per slot leads to taller slots and worsens the field effect (described earlier) in comparison to a single conductor per slot. The coefficient $\alpha$ (in Equation (3)), taking into account the number of conductors per slot, $z_s$, can be calculated using the following analytical relationship [1]:

$$\alpha = \sqrt[4]{\frac{1}{z_s \cdot (z_s - 1) + \frac{4}{15}}} \tag{8}$$

Consequently, according to the Equations (3) and (8) with $z_s$ = 2 and frequency of 1800 Hz, the optimum height of the bar is equal to 1.6 mm. The copper bars are generally cut from standard-sized sheets, so we have chosen the nearest available size of 2 mm.

### 3.2. Simplified Parametric Study

The number of degrees of freedom for the overall optimisation is restricted because the main dimensions (specification) and the fundamental electrical frequency are imposed. Additionally, the bar dimensions and the number of conductors per slot are directly inferred from the frequency and the other electrical parameters. We therefore chose to perform a quasi-optimisation based on a sensitivity study with respect to the few remaining parameters, in particular the slot and magnet dimensions, as shown in Figure 9.

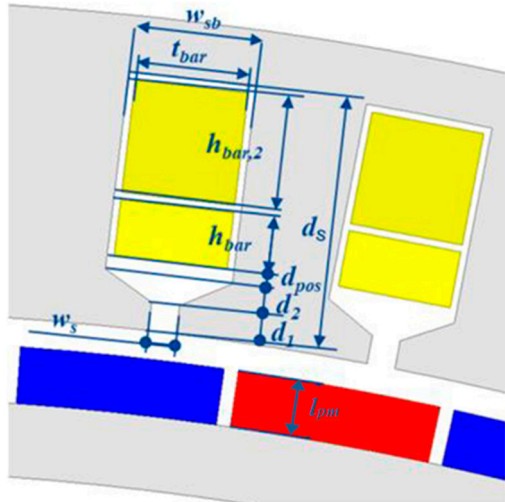

**Figure 9.** Parameters considered in the sensitivity analysis.

In order to perform this sensitivity analysis, the torque (160 Nm), the outer diameter of the stator (240 mm), and the stator stack length (60 mm) have been kept unchanged. The phase current has been adjusted accordingly from the baseline value (200 Arms) in order to maintain the torque (160 Nm).

In the Reference [1], it has been demonstrated that for a configuration with two bars per slot (double layer), it could be interesting, from the point of view of AC copper loss, to modulate the height of the two bars with the upper bar being taller than the bar close to the air gap. Figure 10 shows the evolution of the total copper loss as a function of height $h_{bar2}$ whilst fixing $h_{bar}$ = 2 mm and $t_{bar}$ = 4 mm. Given the sizing constraints with an imposed outer diameter, the optimal value of $h_{bar2}$ remains at 2 mm, which is the height value we have chosen for the two bars.

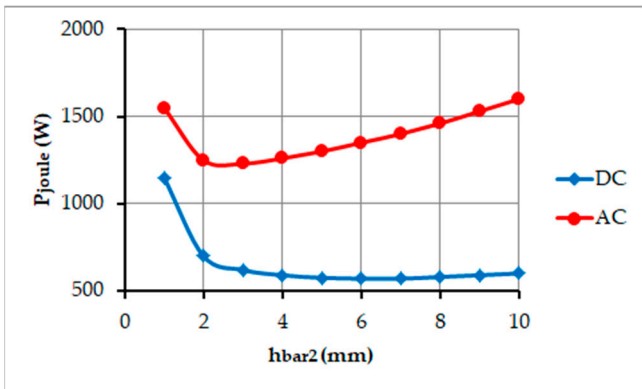

**Figure 10.** DC and AC copper losses versus the height of the upper bar $h_{bar2}$.

The second parameter studied is the magnet thickness, lpm, considering the same constraints. The Figure 11 shows the evolution of the total copper loss as a function of the magnet thickness. It can be seen that the magnet thickness has almost no influence on the losses and thereby the value lpm = 4 mm was chosen to satisfy the requirement of the rotor inertia (which was difficult to fulfil).

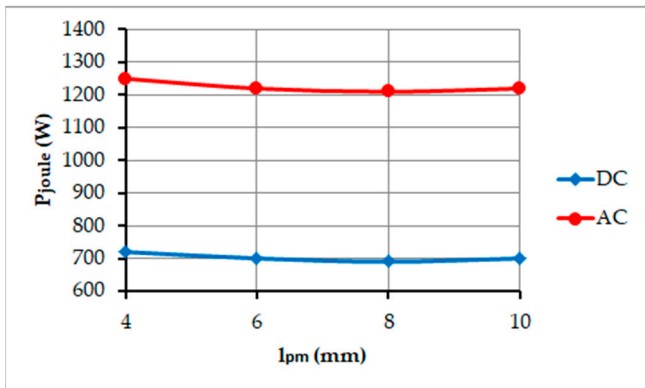

**Figure 11.** DC and AC copper losses versus the magnet thickness $l_{pm}$.

Finally, the last parameter varied is the bar width, $t_{bar}$. Figure 12 shows that the optimum is obtained for $t_{bar}$ = 4.5 mm.

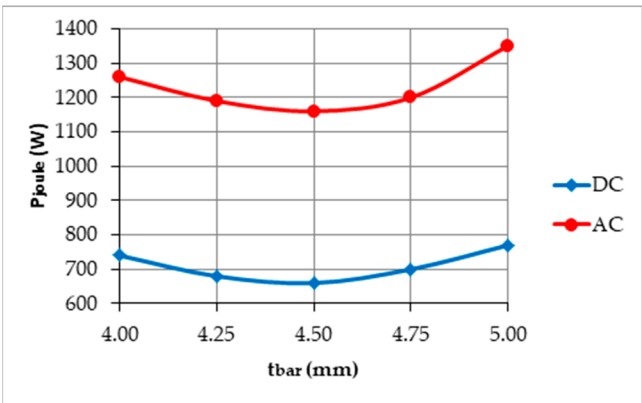

**Figure 12.** DC and AC copper losses versus the bar width $t_{bar}$.

### 3.3. Materials Selection and Dimensions of the Prototype

It is demonstrated in [2] that the use of magnetic materials with a high level of saturation, namely the Iron-Cobalt alloys (FeCo), is not always judicious to reduce the active mass of electrical machines operating at very high frequency, because these or other mechanical considerations which will, in the end, fix the thicknesses of the magnetic back iron/yoke. These mechanical constraints consist of the minimum dimensions required for the cutting of the sheets and for the shrink fitting of the stator into its housing. However, the FeCo offers 20 to 30% lower core losses in comparison with the FeSi alloy at given lamination thickness and operating conditions. It was the commercial grade Vacoflux50®, with sheet thickness of 0.2 mm, that we selected despite its extremely high cost.

The operation at high fundamental frequency (high pole count) leads to the use of narrow magnets (4.5 mm span); the fundamental consequence of this is that the path of the flux under the magnets becomes extremely short. It is thus possible to remove the magnetic yoke under the magnets without impairing the torque too much (the loss is significantly less than 15%) [1,2], whilst benefiting from the non-negligible reduction of both the total mass of the machine and the inertia of the rotor. This approach has been considered and will be further developed here.

The choice of magnet material and grade is also crucial for the design of the motor. The grades of the Samarium-Cobalt (Sm2Co17) are commonly preferred for the aerospace applications due to their high temperature stability, but the most advanced Iron-Neodymium-Boron (NdFeB) grades are also very attractive thanks to their higher remanent induction ($\approx$1.35 T against 1.19 T) and lower cost.

Table 2 gives the steady-state operating temperature of the different active parts of the motor using FeCo or aluminium rotor, and Sm2Co17 or NdFeB (N45UH) magnets. The calculation is carried out using a finite element analysis tool. The heat transfer coefficient (HTC) at the external surface of the motor (forced airflow from the propeller) has been kept unchanged at 200 W/m$^2$K for all the configurations/designs, this coefficient will be validated later. Table 2 shows the required current to reach the output torque of 160 Nm at the start of the simulation with motor temperature of 20 °C; the current differs from one configuration to another.

**Table 2.** Thermal analysis results for the three different designs.

| Configuration | I | II | III |
|---|---|---|---|
| Magnets grade | N45UH | N45UH | Sm$_2$Co$_{17}$ (B$_r$ = 1.1 T) |
| Rotor material | Vacoflux50® | Aluminium | Aluminium |
| T$_{magnet}$ (°C) | 76 | 88 | 124 |
| T$_{winding}$ (°C) | 100 | 122 | 201 |
| T$_{rotor}$ (°C) | 76 | 88 | 124 |
| T$_{stator}$ (°C) | 98 | 120 | 167 |
| Phase current (Arms) | 185 | 225 | 288 |
| Total losses (kW) | 2.2 | 2.75 | 4.56 |
| Efficiency (%) | 94 | 92 | 88 |
| Torque (Nm) | 133 | 128 | 143 |

Figure 13 shows the steady-state temperature distribution within the motor for the configuration III.

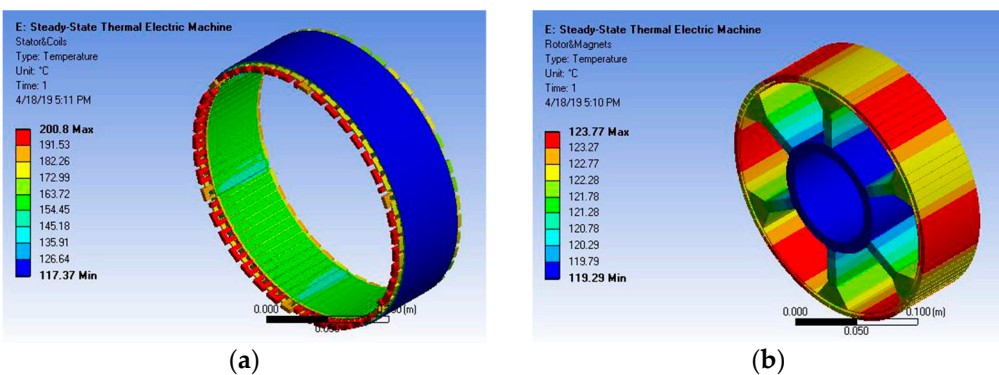

**Figure 13.** Temperature distribution for the configuration III: (**a**) stator, (**b**) rotor.

According to the results of this study, it appears that the motor performances are lower with a non-magnetic rotor due to the excessive steady-state temperature in the different active parts of the motor. Moreover, despite their higher sensitivity to temperature, the use of NdFeB magnets gives better results (lower temperature).

Table 3 summarises the continuous performance of the selected design for the prototype based on the configuration I.

**Table 3.** Characteristics of the selected design (Configuration I).

| Performance | | Steady-State Temperatures | |
|---|---|---|---|
| Phase current (Arms) | 253 | $T_{magnet}$ (°C) | 108 |
| Back EMF constant Ke (Vs/rad) @ 20 °C | 0.29 | $T_{winding}$ (°C) | 152 |
| Total losses (kW) | 4.2 | $T_{rotor}$ (°C) | 112 |
| Efficiency (%) | 90 | $T_{stator}$ (°C) | 148 |
| Torque (Nm) | 160 | | |

The temperature rise in the magnets is caused by the eddy-current losses and mainly the loss component due to the armature reaction (stator excitation) [1]. For the selected configuration I, the eddy-current losses in the magnets are around 100 W when considering solid magnets. In order to reduce the losses, the magnets have been segmented in the axial direction as illustrated in Figure 14.

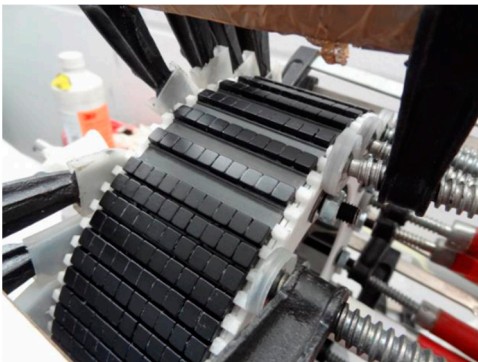

**Figure 14.** Axial segmentation of the magnets.

The analysis of the magnet eddy-current loss as a function of the number of segments is presented in Figure 15 [1]. It can be seen that a loss reduction of approximately 30 to 40% can be achieved by sufficiently segmenting the magnets in the axial direction; however, this to the detriment of the manufacturing cost. The results in Tables 2 and 3 take into account this loss reduction.

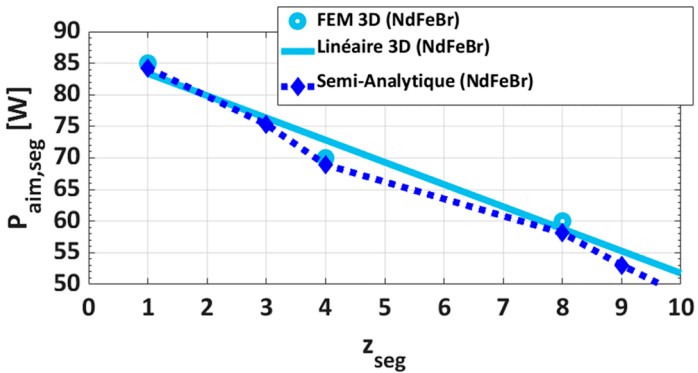

**Figure 15.** Evolution of the magnets eddy-current loss as a function of the axial segmentation.

Even though the final design presented in Table 3 is globally close to the objective dictated by the specifications, the efficiency remains slightly below the targeted value. The only solution to remedy this is to optimise the cooling via the housing fins placed on the periphery of the motor, as illustrated in Figure 16.

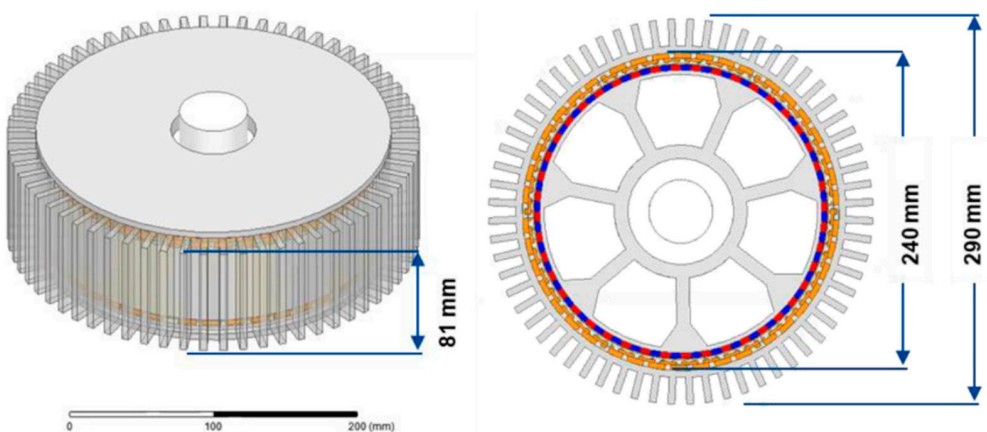

**Figure 16.** Study of the housing fins for the cooling.

Figure 17 shows the evolution of the HTC (convection) as a function of the required fan power, which is dictated by the number and arrangement of the fins. The result was obtained from a simple analytical model of fins [1] considering a fin height of 16.5 mm. Despite its inaccuracy, the analytical model validates the cooling assumptions and shows that the HTC initially chosen (200 W/m$^2$ K) for the calculation of the motor performance in Tables 2 and 3 is rather pessimistic.

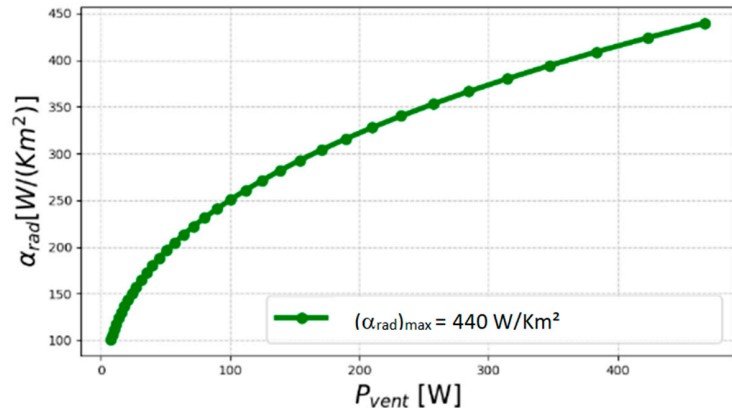

**Figure 17.** HTC as a function of the fan power.

In order to corroborate this analytical result, we cross-checked it by using a tool numerically solving the fluid mechanics equations. Figure 18 represents the convection coefficient along a fin, for an air inlet speed of 24 m/s. The average HTC along the fin is equal to 280 W/m$^2$ K.

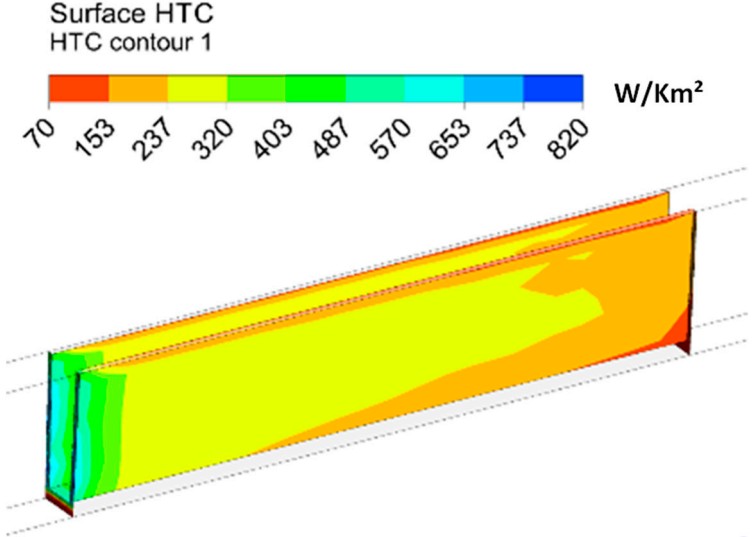

**Figure 18.** Simulated airflow along the fin.

In the end, for the manufacturing of the prototype, we chose a configuration with staggered fins, as illustrated in Figure 19, which improves the heat transfer by about 20% by homogenising the temperature of the fins.

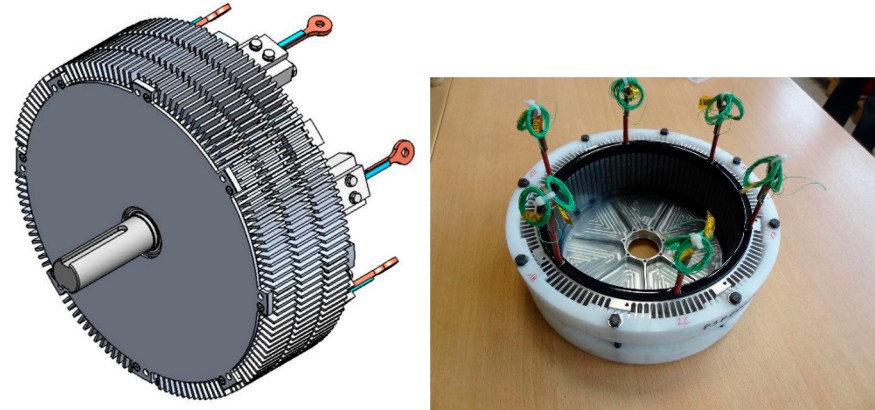

**Figure 19.** View of the complete prototype.

## 4. Experimental Test Results, Performance of the Prototype and Conclusions

The Table 4 presents the main characteristics of the prototype.

**Table 4.** Main characteristics of the prototype.

| | |
|---|---|
| Nominal output torque | 160 NM |
| Maximum continuous power @ 2500 RPM | 42 kW |
| Nominal efficiency | 90% |
| Overall outer diameter | 290 mm |
| Total length | 92 mm |
| Total mass/active mass | 9.7 kg/5.7 kg |
| Power-to-weight ratio, total/active | 4.3 kW/kg/7.4 kW/kg |
| Torque-to-weight ratio, total/active | 16.5 Nm/kg/28 Nm/kg |

To date, the on-load test campaign has not been completed yet, but the no-load and short-circuit tests have given the following results summarised in Table 5.

**Table 5.** Experimental test results.

| | |
|---|---|
| Back EMF constant Ke (phase to neutral RMS) @ 20 °C | 0.26 Vs/rd |
| Torque constant Kt @ 20 °C | 0.78 Nm/Arms |
| Phase resistance @ 100 °C | 12.4 mΩ |
| Phase inductance | 16.4 μH |
| No-load core losses @ 2500 rpm | 900 W |

The efficiency of the motor calculated based on the parameters in Table 5 is slightly higher than 90%, but this value is optimistic because the core losses measured at no-load are not fully representative of the total core losses in the stator core pack under full load. However, these preliminary measurements are in good agreement with the results from the theoretical sizing.

In conclusion, the performance of the prototype is quite close to that defined by the specifications; only the efficiency of 92% cannot be fully achieved due to the envelope constraints. The continuous power-to-weight ratio obtained is outstanding (cf. Table 4); taking into account the relatively low speed of rotation, it is certainly almost impossible to achieve better than 5 kW/kg (overall & continuous) with this motor topology. The continuous torque-to-weight ratio is also exceptional.

**Author Contributions:** Conceptualization, D.M., L.P., A.G., P.E. and M.A.; investigation, L.P.; methodology, L.P.; writing—original draft, D.M.; writing—review and editing, N.B. All authors have read and agreed to the published version of the manuscript.

**Funding:** This research was funded by SAFRAN TECH from SAFRAN Group.

**Institutional Review Board Statement:** Not applicable.

**Informed Consent Statement:** Not applicable.

**Data Availability Statement:** Not applicable.

**Conflicts of Interest:** The authors declare no conflict of interest.

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
