# Peer review of "Low-Voltage, High-Frequency Synchronous Motor for Aerospace Applications"

_electronics, doi:10.3390/electronics11172719_

Round 1

Reviewer 1 Report

This paper studies a PM electric motor used for driving propeller of aircrafts, which uses solid bar winding instead of traditional round wire winding. Especially,the optimization of the slot,the copper conductor, permanent magnet size other design parameters is presented in the paper. Finally, the main experimental test results of the prototype are shown and analyzed. The performance of the prototype is in a good agreement with the design. The optimization design of the PM electric motor, especially the solid bar winding design, is interesting and some publishable results are contained.

 Major comments

1. The Abstract Section should be rewritten, because the focuses and innovation of this paper, in the abstract, are not expressed. Abstract should be concise.

 2. In the Abstract Section, authors write that “It has been proven that…a continuous power-to-weight close to 10 kW/kg”, but in Section 4, authors write that “it is certainly almost impossible to achieve better than 5 kW/kg …”. So, what is the power-to-weight ratio of the motor in this paper? It should need to be further determined.

Minor comments

1. Figures in the paper should be standardized, in the reviewer’s opinion. Including the line thickness of the curves, the size and type of fonts in Figures, the specification of the coordinate axes, etc. For example, Figure 10, 11, and 12 should be redrawn.

Author Response

Responses in the attached document.

Reviewer 2 Report

This paper is very interesting and clear demonstrates the potential benefits to have better power-to-weight ratio. The followings are the reviewer's concern to this paper.

The literature review and discussion are missing. It is recommended to provide a comprehensive literature review including the gap in the previous studies and summary of the reviewer's perspective. 

Line 41. If the authors need the citation or resource of pictures, please include them.

Line 50. What is the PM?

Line 58. Please make sure that table 1 is fitted in page 2.

Author Response

Responses in the attached document
